# *MICAL1* Monooxygenase in Autosomal Dominant Lateral Temporal Epilepsy: Role in Cytoskeletal Regulation and Relation to Cancer

**DOI:** 10.3390/genes13050715

**Published:** 2022-04-19

**Authors:** Sipan Haikazian, Michael F. Olson

**Affiliations:** Department of Chemistry and Biology, Ryerson University, Toronto, ON M5B 2K3, Canada; sipan.haikazian@ryerson.ca

**Keywords:** autosomal-dominant lateral temporal epilepsy, *LGI1*, *RELN*, *MICAL1*, semaphorin signaling, axonal development, cancer genomics

## Abstract

Autosomal dominant lateral temporal epilepsy (ADLTE) is a genetic focal epilepsy associated with mutations in the *LGI1*, *RELN,* and *MICAL1* genes. A previous study linking ADLTE with two *MICAL1* mutations that resulted in the substitution of a highly conserved glycine residue for serine (G150S) or a frameshift mutation that swapped the last three C-terminal amino acids for 59 extra residues (A1065fs) concluded that the mutations increased enzymatic activity and promoted cell contraction. The roles of the Molecule Interacting with CasL 1 (MICAL1) protein in tightly regulated semaphorin signaling pathways suggest that activating MICAL1 mutations could result in defects in axonal guidance during neuronal development. Further studies would help to illuminate the causal relationships of these point mutations with ADLTE. In this review, we discuss the proposed pathogenesis caused by mutations in these three genes, with a particular emphasis on the G150S point mutation discovered in *MICAL1*. We also consider whether these types of activating *MICAL1* mutations could be linked to cancer.

## 1. Clinical Characteristics and Genetic Transmission of ADLTE

Autosomal dominant lateral temporal epilepsy (ADLTE), also known as autosomal dominant epilepsy with auditory features, is a form of genetic focal epilepsy characterized by seizures affecting one hemisphere of the brain [1,2]. ADLTE onset is associated with mutations in the *LGI1*, *RELN*, or *MICAL1* genes that code for the Leucine-rich Glioma-Inactivated 1 (LGI1), Reelin, and the molecule interacting with CasL 1 (MICAL1) proteins, respectively [3,4,5]. ADLTE is transmitted in an autosomal-dominant fashion, in which only one dominant acting allele is required for the phenotype to be manifested [6]. This disease typically presents in patients as auditory symptoms and aphasia, and inabilities with language comprehension, despite normal computed tomography (CT) and magnetic resonance imaging (MRI) results [7]. These presentations are usually required to establish a diagnosis, although clinicians may perform genetic testing for the previously mentioned genes if symptoms are inconsistent [7].

The typical developmental stages of onset are in adolescence and early adulthood, with an unknown but estimated very low prevalence within the population. In general, it has been estimated that epilepsies transmitted in a Mendelian manner, including ADLTE, account for a small percentage of all individuals living with the condition [8]. Much of what is known about ADLTE has been discovered through small-scale family studies with individuals either suspected or confirmed to have the disease, dating back to studies conducted in the mid-1990s [9]. Given the debilitating symptoms for those individuals living with ADLTE, more basic science and clinical research is required to understand the etiologies and molecular pathogenesis of the disease.

## 2. *LGI1*, *Reelin* and *MICAL1* Genes in the Etiology of ADLTE

### 2.1. LGI1

The *LGI1* gene is located on chromosome 10 band 10q23.33, and LGI1 is a secreted glycoprotein expressed in neurons [10]. LGI1 has been hypothesized to regulate the expression of Kv1 potassium channels through its interactions with membrane-anchoring proteins such as members of the “a disintegrin and metalloproteinase” (ADAM) family including ADAM22, and the postsynaptic membrane-associated guanylate kinase (MAGUK) protein family, thus ensuring normal neuronal signaling [11,12]. LGI1 forms a heterotetramer with ADAM22 in neuro, and this protein complex has been suggested to regulate receptors such as α-amino-3-hydroxyl-5-methyl-4-isoxazolepropionic acid (AMPA) and N-methyl-D-aspartic acid (NMDA) receptors [13,14]. In addition, Kv1 channels responsible for neuronal repolarization were found to be positively regulated by LGI1-ADAM22 complexes [15]. Two groups of LGI1 mutations have been proposed to affect these interactions, classified either as secretion-defective or secretion-competent [16]. In secretion-defective mutations, misfolded protein products are degraded by the endoplasmic reticulum and are therefore unable to bind to ADAM22 [13]. Disruptions in electrical currents may occur as a result. In secretion-competent mutations, the protein is secreted, but mutations render it incapable of binding ADAM22 [13]. However, there are other secretion-competent mutations that result in binding to ADAM22 but failure to form heterotetramers [17]. In any case, defective LGI1-ADAM22 complexes can disrupt Kv1-mediated currents, resulting in a failure to inactivate the neuron following depolarization [12,13] and consequent neuronal overexcitability, which ultimately may be responsible for the epilepsy symptoms observed in ADLTE (Figure 1a,b) [12,14].

Mutant LGI1-mediated alterations in glutamatergic synapse development have also been proposed as a causative mechanism of ADLTE [18]. During the postnatal period, neurons downregulate the release of glutamate from presynaptic neurons and reduce expression of on postsynaptic neurons. LGI1 was shown to play a key role in this NMDA receptor downregulation [18]. Transgenic mice harboring a truncated *LGI1* mutation (835delC) implicated in ADLTE showed increased presynaptic glutamate release and NMDA-receptor expression in the neurons of the hippocampal dentate gyrus compared to mice with the wild-type form of the protein [18]. The lower epileptic threshold of patients with ADLTE could result from the increased activation of excitation circuitry that remains throughout their lives [19]. 

Over 40 ADLTE-causing *LGI1* mutations have been identified, with most mutations inhibiting the secretion of the protein into the extracellular matrix [17,20]. However, some studies have also found secretion-independent pathogenic mechanisms, leaving open the possibility of other mechanisms being responsible for the molecular pathogenesis [21]. The ADLTE-associated *LGI1* mutation accounts for 67% of patients manifesting symptoms of the disease, and an estimated 30% of total ADLTE diagnoses are attributable to *LGI1* mutations [1,22]. Current research on the involvement of LGI1 in ADLTE is largely focused on identifying and characterizing new mutations [23]. 

### 2.2. RELN

The *RELN* gene is located on chromosome 7 band q22.1, and the Reelin gene product is a secreted glycoprotein that contributes to the proper migration of cells in parts of the embryonic forebrain during embryogenesis, and axonal, dendritic and synapse development in the adult brain [24]. ADLTE patients with *RELN* mutations were found to have lower serum levels of the protein than those without the mutation [4]. Clarification of a potential relationship between the amount of Reelin and the pathogenesis of ADLTE came from a follow-up study that found that low levels of Reelin in the blood and extracellular matrix in the brain could lead to epilepsy [25], summarized in Figure 2. However, the precise mechanisms by which mutant Reelin might cause ADLTE have yet to be elucidated. 

Pathogenic mutations in the *RELN* gene are thought to be causal in 17–18% of ADLTE diagnoses [1]. Since research has only recently implicated *RELN* in ADLTE, the penetrance of the mutations is unknown [1]. In a study that included 33 ADLTE-diagnosed patients, 60% of individuals showed a phenotype characteristic of ADLTE, consistent with the mutation having relatively high penetrance [4]. Additional research is needed to understand the pathogenic mechanism of mutant Reelin.

### 2.3. MICAL1

Isolated genetic case reports have suggested that unknown mutant genes at various chromosomal loci are linked to ADLTE-like symptoms and seizures [26,27]. An estimated 50% of ADLTE cases could be due to mutations of these uncharacterized genes [7]. Mutations of *MICAL1*, located on chromosome 6 band q21, were recently found to be linked to ADLTE in a cohort of Italian families [3,10]. In two families, researchers identified two different mutations to *MICAL1*: a glycine to serine substitution at the 150th amino acid (G150S), and a frameshift deletion mutation at the 1065th alanine (A1065fs) that resulted in the deletion of the last three C-terminal amino acids and the addition of 59 extra residues [3]. A more thorough review of *MICAL1* and its associated protein product will be provided in the following section to further understand the significance of these pathogenic variants.

## 3. Structure and Function of MICAL1

MICAL1 is an intracellular multi-domain enzyme, a member of the MICAL family of proteins involved in the regulation of filamentous actin (F-actin) dynamics [28]. The other closely related members of the human MICAL family are MICAL2 and MICAL3, with each corresponding gene being located on chromosome 11 band p15.3 and 22 band q11.21, respectively [29]. There are two additional members of the MICAL family that lack catalytic activity, MICAL-L1 and MICAL-L2, which will not be discussed further in this review. MICAL2 and MICAL3 share a similar function with MICAL1 in F-actin disassembly but interact with the actin cytoskeleton in different ways, contributing to different morphological changes in cells [29]. In HeLa cells, ectopic expression of MICAL1 was shown to directly reduce levels of F-actin, whereas MICAL2 expression led to structural changes to F-actin, resulting in filopodial-like structures [30]. siRNA-mediated knockdown of each protein also had different effects on cells; actin protrusions were seen in cells treated with MICAL1 or MICAL2 siRNA, while increases in cell size were noted in MICAL3 knockdown cells [30]. While the MICAL family of proteins are expressed in the nervous system and have been found to play key roles in the developing nervous system, expression levels differ throughout development [31]. MICAL1 and MICAL3 are widely expressed in the adult rat nervous system, whereas MICAL2 expression is largely absent in areas of the central nervous system such as the basal ganglia and the hypothalamus [31].

The MICAL1-3 proteins share similar domains, with each containing monooxygenase (MO), calponin-homology (CH), and Lin-11, Isl-1, and Mec-3 (LIM) domains [32]. The percentages of amino acid identity between the three MICAL proteins and between their domains are depicted in Figure 3. MICAL1 and MICAL3 also contain a Rab-binding domain, also called a C-terminal coiled-coil (CC) domain, which is capable of autoinhibition of enzymatic activity [27,30]. Another major difference between these proteins is their size; MICAL1 is composed of 1067 amino acids, and MICAL2 is made of 1124 amino acids, whereas MICAL3 consists of 2002 residues due to a large addition between the LIM and RBD segments [27,30]. Interestingly, the degree of amino acid homology suggests that MICAL2 and MICAL3 are more closely related to each other than to MICAL1, although MICAL2 is exceptional due to its lack of the the RBD.

Through its enzymatic MO domain, MICAL1 catalyzes oxidation-reduction (redox) reactions through a monooxygenase or oxidase mechanism, which is dependent on an interaction with the 4′-OOH molecule on the flavin adenine dinucleotide (FAD) cofactor [33]. MICAL1 undergoes a two-step redox reaction when interacting with F-actin. An oxidized flavin ring in FAD reacts with NADPH to form reduced flavin and NADP+. The reduced flavin then reacts with oxygen to form the intermediate 4a-hydroperoxy-flavin, which then produces oxidized flavin and hydrogen peroxide (H_2_O_2_) [34,35]. Through this oxidase activity, the H_2_O_2_ produced and the high local concentration of F-actin results in the stereoselective oxidation of methionine residues, leading to F-actin disassembly into monomeric globular actin (Figure 4) [36]. The possibility remains that the diffusible nature of H_2_O_2_ produced by active MICAL1 could also result in the oxidation of additional proteins associated with or proximal to the MO domain.

In support of this proposed mechanism of action, a study found that direct binding of F-actin to MICAL1 increased its catalytic activity, resulting in the reaction of H_2_O_2_ with numerous methionine and tryptophan residues on F-actin, and consequent depolymerization [37]. Although actin disassembly due to H_2_O_2_ diffusion is a plausible reaction mechanism, other mechanisms for MICAL1-mediated actin oxidation have been proposed [35]. One theory is that MICAL1 directly binds F-actin and stereospecifically adds oxygen directly to methionine residues, specifically Met44 and Met47, leading to the formation of methionine R-sulfoxide [38]. This biochemical mechanism is also supported by studies on MICAL2 and MICAL3 that made similar conclusions [32].

The current literature does not clarify the precise catalytic mechanism of the enzyme [35]. Several studies have suggested that the MICAL1 MO domain undergoes multiple conformational changes to carry out its catalytic activity, similar to other flavin-dependent monooxygenases [34,39]. As the first reaction in the catalytic mechanism is proposed to be NADPH oxidation and subsequent FAD reduction, looking at the residues involved in facilitating this reaction could provide insights into how the overall mechanism may be carried out [40]. Structural and functional analysis suggest that a catalytic loop consisting of amino acids 395 to 405 (Figure 5) is involved in mediating this first step [41]. Conformational changes to the tryptophan at position 400 (W400), which is involved in pi stacking with the aromatic rings of the bound FAD to stabilize binding [40], could be involved in transferring a hydride from NADPH to FAD [35,41]. Using crystallographic structures of mouse MICAL1, researchers proposed that actin methionine residues are near W400 during the catalytic cycle, making them highly accessible to the reactive H_2_O_2_ [35,37].

The other domains of MICAL1 also play important roles in facilitating and regulating the oxidase activity of the enzyme. The CH domain in full-length MICAL1 has been considered to be important for facilitating the interaction between the MICAL1 MO domain and actin [42]. More recently, a model was proposed in which intermolecular interactions between the MO domain of one MICAL1 protein and the CH domain of another MICAL1 protein can occur, increasing binding between MICAL1 and F-actin [41].

Structural studies support the theory of the CH domain playing a facilitating role in enzyme activity as the isolated CH domain was not found to bind actin [43]. Another study found that both domains may be involved in the kinetics of NADPH binding to MICAL1 via changes in local pH [37]. LIM domains are commonly found in proteins involved in cytoskeletal dynamics [44], suggesting that they may contribute to MICAL1 localization. The LIM domain may also be involved in autoinhibiting the catalytic domain and binding to other proteins in the cytoskeleton [45]. However, it has not yet been established whether the CH and LIM domains play roles in regulating the catalytic domain by binding other interacting proteins. The C-terminal domain has been suggested to play an autoinhibitory role [37], rendering the MICAL1 holoenzyme protein functionally inactive [25]. In vitro studies found that the removal of the C-terminal region resulted in constitutive activation of the enzymatic domain [45]. Due to the possibility of MICAL1 existing in multiple protein conformations, there may be an equilibrium between active and inactive MICAL1 conformations that could be shifted in one direction or the other by factors such as protein binding or post-translational modifications [37].

The MICAL1 C-terminus interacts with cytoplasmic plexin/semaphorin complexes that are involved in the axonal growth of developing neurons, which could result in relief of autoinhibition of the MO domain [37,45]. Understanding the multi-domain structure of MICAL1 is essential for determining the biochemical means by which mutations lead to ADLTE. The alanine frameshift mutation (A1065fs) alters the composition of the C-terminal domain by deleting the terminal three amino acids and adding 59 residues, which could alter the conformation of the C-terminal region and interrupt the autoinhibitory role that the domain plays, resulting in increased enzymatic activity [3]. 

## 4. Biochemical Roles of G150 and A1065

The G150 amino acid is located in the MICAL1 MO domain, has identically positioned homologous glycine residues in human and mouse MICAL2 and MICAL3, and is highly conserved in MICAL1 across multiple species, suggesting that it is important for enzyme function [3]. The A1065 residue is the third last, normally followed only by Q1066 and G1067 residues. The expression of mutant G150S or A1065fs MICAL1 in HEK293T human embryonic kidney cells resulted in greater levels of oxidoreductase activity relative to wild-type MICAL1, consistent with the mutations increasing enzymatic specific activity [3]. Interestingly, there was a more pronounced increase in activity for human G150S MICAL1 compared to the mouse version with the same mutation, suggesting that there are additional factors that affect the outcome of the amino acid change [3]. In addition, the increase in enzyme activity for the A1065fs mutation was greater than for the G150S mutation, consistent with the C-terminus exerting significant autoinhibition that is relieved by the A1065fs mutation. The expression of human MICAL1 G150S and A1065fs mutants in COS-7 African green monkey kidney cells induced significant and comparable cell contraction relative to the wild-type protein, while the G150S mutation in mouse MICAL1 did not produce significant effects [3]. The observed effects on cell morphology were proposed to result from the increased oxidoreductase activity that would be predicted to result in actin depolymerization [3,45]. 

While the presumed mechanism of action of the C-terminal A1065fs mutation is straightforward, the position of G150 in the protein does not make it clear how the G150S mutation would increase catalytic activity. An unpublished crystal structure of mouse MICAL1 (PDB 4TXK) reveals G150 to be located on a loop that positions the amino acid on the protein surface directly opposite from the pocket in which FAD sits (Figure 6). The homologous G152 residue in MICAL3 is similarly positioned, consistent with the importance of this residue for normal enzyme activity [46]. Interestingly, a published crystal structure of mouse MICAL1 did not resolve the position of G150, suggesting that the loop in which it is positioned may be relatively unstructured and flexible [42]. The study by Dazzo et al. that identified the *MICAL1* mutations associated with ADLTE hypothesized that the G150S mutation might alter the conformation of the MO domain such that the autoinhibitory actions of the C-terminal domain would be diminished [3]. However, this model has not been biochemically substantiated. Given the limited amount of information about the MICAL1 mechanism of catalysis and the apparent distance of G150 from the conserved catalytic loop that includes W400 (Figure 5 and Figure 6), further investigation is warranted. Examining the functional interactions between G150 and W400 could help determine the biochemical means by which the G150S substitution increases enzymatic activity. For example, a particular rotamer of WT MICAL1 may involve G150 interacting with the catalytic loop to facilitate hydride transfer. Mutations of glycine into the polar serine residue may influence the polarity of interactions in the catalytic loop, thereby allowing for a greater degree of FAD-binding and subsequent oxidase activity to occur. An additional possibility is that the G150S substitution creates a potential site of phosphorylation that could directly affect the MO domain to increase activity or might result in reduced autoinhibition by the C-terminal domain due to steric hindrance or charge repulsion. The determination of the structure of the MICAL1 holoprotein would illuminate the mechanism of autoinhibition by the C-terminus and could reveal how the G150S mutation increased catalytic activity. 

## 5. MICAL1 in Central Nervous System Biology and ADLTE Etiology

The precise molecular pathogenesis of ADLTE is unknown. The current literature suggests that dysregulated neuronal development is an important contributory factor [7]. MICAL1 has a vital role in the axonal guidance of neurons, the process by which axons grow along well-defined paths in the nervous system [28]. One way that MICAL1 has been implicated in this process is through its interaction with semaphorin signaling pathways. Axons grow using neuronal growth cones, extensions of developing neurons mediated by actin assembly and disassembly [47]. The path that growth cones take is dictated by numerous guidance molecules, including semaphorins. These are a class of membrane-bound, GPI-anchored, and secreted proteins that exert repulsive effects on axonal guidance, preventing axons from growing into specific areas [48]. Semaphorins accomplish this task by binding to their plexin transmembrane receptors, which are located on the surface of growth cones [49]. MICAL1 was proposed to be involved in this signaling modality since the C-terminus of *Drosophila melanogaster* MICAL was found to bind to the plexin PlexA, leading to disassembly of actin filaments and changes in axon movement [28]. MICAL1 may also indirectly affect microtubule dynamics, which play a structural role in developing growth cones [50] since the H_2_O_2_ produced by the MICAL1 MO domain oxidized the collapsing response mediator protein 2 (CRMP2), a protein involved in assembling microtubules [51,52]. The CRMP2 oxidation resulted in the formation of a transient complex with oxidoreductase thioredoxin, which inhibited microtubule assembly and led to growth cone collapse, the process by which cells can arrest their current outgrowth [51]. Thus, growth cone morphology and activity in axon guidance are influenced by the enzymatic activity of MICAL1.

Given the involvement of MICAL1 in semaphorin signaling, mutations that affect actin filament disassembly may disrupt the tightly regulated movement and stabilization of the growing axon during development, leading to the symptoms seen in ADLTE [3]. The increased cell contraction induced by mutant MICAL1 compared to the wild-type protein is consistent with dysregulated growth cone collapse leading to ADLTE onset [3]. The COS7 cell contraction assay used to evaluate the effect of the *MICAL1* mutations has been suggested to be predictive of neuronal growth cones collapse [53,54]. Therefore, increased cell contraction suggests an increased ability of the cell to collapse the growth cone and subsequently misguide axons, thus contributing to ADLTE. Figure 7 summarizes the hypothesized role MICAL1 plays in the molecular mechanism of growth cone collapse. Support for actin dysregulation being a causative mechanism for ADLTE can also be seen in the function of other proteins implicated in the disease, including Reelin [4,55]. One cellular function of Reelin is the inactivation of the F-actin severing protein Cofilin, consistent with the potential links between F-actin regulation and ADLTE [55]. Despite the mechanisms discussed, the effects of ADLTE-associated MICAL1 mutations on F-actin structures and cytoskeletal dynamics have not been directly studied. 

## 6. ADLTE-Associated *MICAL1* Mutations in Cancer

While the most well-characterized functions of MICAL1 are associated with regulation of the organization of the central nervous system, it has been implicated in several different cancers, including breast, gastric, colorectal, and melanoma [56,57,58,59]. As a regulator of the actin cytoskeleton, it has been most well studied in the context of metastasis, the process by which cancer cells disseminate from primary tumors and move to distant tissues within the body [60]. The cytoskeleton plays important roles in the process of metastasis through its cycles of actin remodeling driven, in part, by actin polymerization and depolymerization [61]. Therefore, metastasis-promoting mutations in cytoskeletal regulators could be those that increase or decrease enzymatic activity relative to the wild-type protein, depending on the regulatory role of the specific protein and the context in which the mutations occur. Furthermore, high levels of reactive oxygen species (ROS), such as H_2_O_2_, have been associated with numerous cancer types [62]. In sufficiently high quantities, ROS may interact with a number of signaling pathways to promote uncontrolled cell proliferation [62]. In terms of metastasis, numerous in vitro studies using various cancer cells lines have shown that increased ROS is associated with properties associated with enhanced metastasis, including increased motility and invasiveness [63,64]. Given that H_2_O_2_ is produced by MICAL1, increased MICAL1 activity resulting from point mutations such as G150S truncating mutations, or mutations such as A1065fs that disrupt the autoinhibitory actions of the C-terminal region, could be tumor-promoting [37].

MICAL1 overexpression has been associated with elevated ROS generation, which contributed to the increased invasiveness of a number of human breast cancer cell lines [57]. Furthermore, the depletion of *MICAL1* in *BRAFV600E*-expressing melanoma cells led to an increase in cell apoptosis [59]. *MICAL1* gene disruption in MDA MB 231 human breast cancer cells was also associated with F-actin rearrangements, impaired directional cell migration, and reduced xenograft tumor growth [56]. Thus, increased MICAL1 activity could be involved in cancer progression through its effects on the cytoskeleton and increased ROS production.

The most common sites of MICAL1 point mutations found in over 90,000 tumor samples from 202 studies in the cBioPortal curated set of non-redundant studies are depicted in Figure 8 [65,66]. Of the mutations in the MO domain observed at least three times, R98 (R98W or R98Q) and R230 (R230Q or R230stop) were the amino acids in which mutations were the most frequent. In the absence of biochemical data, the effect of the amino acid substitutions is unknown, while the truncating R230stop mutation is likely to be inactivating. Interestingly, one G150S point mutation was detected in an endometrial carcinoma tumor sample, suggesting that MICAL1 activation could be tumor-promoting in this case [60]. Mutations outside of the catalytic MO domain, including the more frequent mutations at A758 and E881, could be activating due to the relief of autoinhibition by the C-terminus. Of 41 observed truncating mutations, 22 were located outside of the MO domain between L443 to D1045, any or all of which could be activating due to removal of the C-terminus. The effect of these mutations on MICAL1 activity would have to be characterized in order to determine whether MICAL1 gain or loss of function is the predominant consequence of cancer-associated mutations. Copy number variations were detected in 373 tumor samples, of which 180 were amplifications and 193 were deep deletions, suggesting that there may be cancer-specific contexts that favor increased or decreased MICAL1 activity [65,66].

## 7. Conclusions

ADLTE-associated mutations have been identified in the *LGI1*, *RELN,* and *MICAL1* genes. Although not directly involved in the same biological functions, all three genes likely affect neuronal development and function, consistent with dysfunction of the central nervous system being the etiological basis of ADLTE. Although relatively infrequent, the two *MICAL* mutations, G150S and A1065fs, both result in increased MICAL1 activity. The mechanistic basis of the A1065fs mutation is likely due to relief from autoinhibition imposed by the C-terminal region on the catalytic MO domain. For the G150S mutation, it is not clear how MICAL1 activity is increased. One possibility is similar relief from autoinhibition by the protein C-terminus, mediated in this case by a mutation within the MO domain. Alternatively, the G150S mutation might directly affect the specific activity of the catalytic domain. The consequence of either MICAL1 mutation is likely to have comparable effects on F-actin stability, which subsequently could affect axon guidance during neuronal development. An examination of cancer-associated *MICAL1* mutations suggests that there may be both activating and inactivating mutations, although the biochemical characterization of the consequences of these mutations is lacking. Since an important function of MICAL1 is F-actin depolymerization, and there are many biological processes that depend on the actin cytoskeleton, there may be specific situations or cell types in which either increased or decreased MICAL1 activity is tumor-promoting.

## Figures and Tables

**Figure 1 genes-13-00715-f001:**
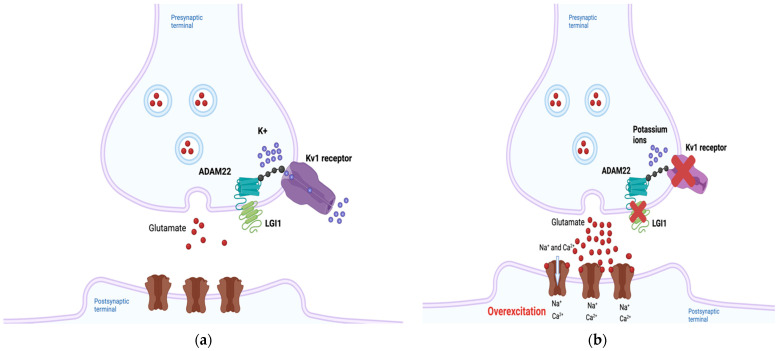
The predicted pathogenic mechanism of Autosomal Dominant Lateral Temporal Epilepsy (ADLTE)-associated *LGI1* mutants. (**a**) Leucine-rich Glioma-Inactivated 1 (LGI1) normally regulates neuronal currents through its indirect association with potassium channels. (**b**) Through several different mechanisms, mutant LGI1 is unable to decrease Kv1 currents, inhibiting repolarization and subsequently increasing glutamate secretion, overexciting the postsynaptic neuron. Created with BioRender.com (accessed on 27 February 2022).

**Figure 2 genes-13-00715-f002:**
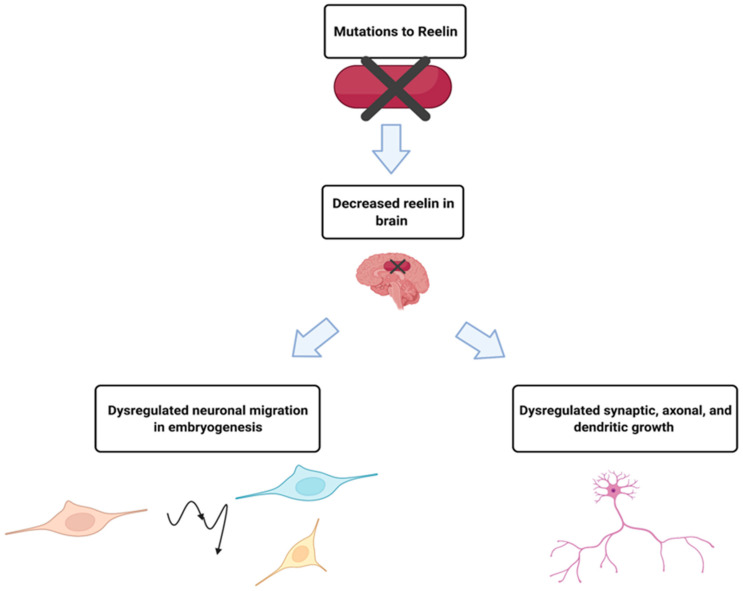
While much is unknown about the mechanism by which mutant Reelin leads to ADLTE, decreased serum levels in ADLTE patients suggested causal roles. It is hypothesized that Reelin is also decreased in the brain of these patients, affecting processes such as neuronal migration, synaptic potentiation, and dendritic growth, which could result in the development of ADLTE. Created with BioRender.com (accessed on 27 February 2022).

**Figure 3 genes-13-00715-f003:**
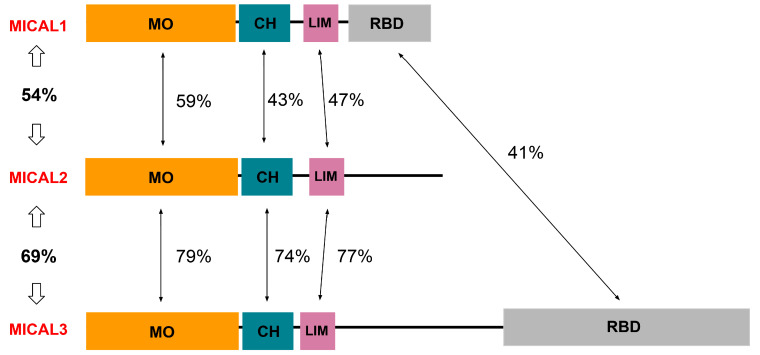
Domain structure and amino acid identity between Molecule Interacting with CasL 1 (MICAL)1–3 family proteins. MO = monooxygenase, CH = Calponin Homology, LIM = Lin-11, Isl-1, and Mec-3 domain, and RBD = Rab-binding domain. Sequence and domain information was from UniProt (https://www.uniprot.org, accessed on 27 February 2022). The Basic Local Alignment Search Tool (BLAST; http://blast.ncbi.nlm.nih.gov/Blast.cgi; accessed on 27 February 2022) was used to calculate similarities between human MICAL1 (NCBI Reference Sequence: NP_001152763.1), human MICAL2 (NP_001269592.1), and human MICAL3 (NP_056056.2).

**Figure 4 genes-13-00715-f004:**
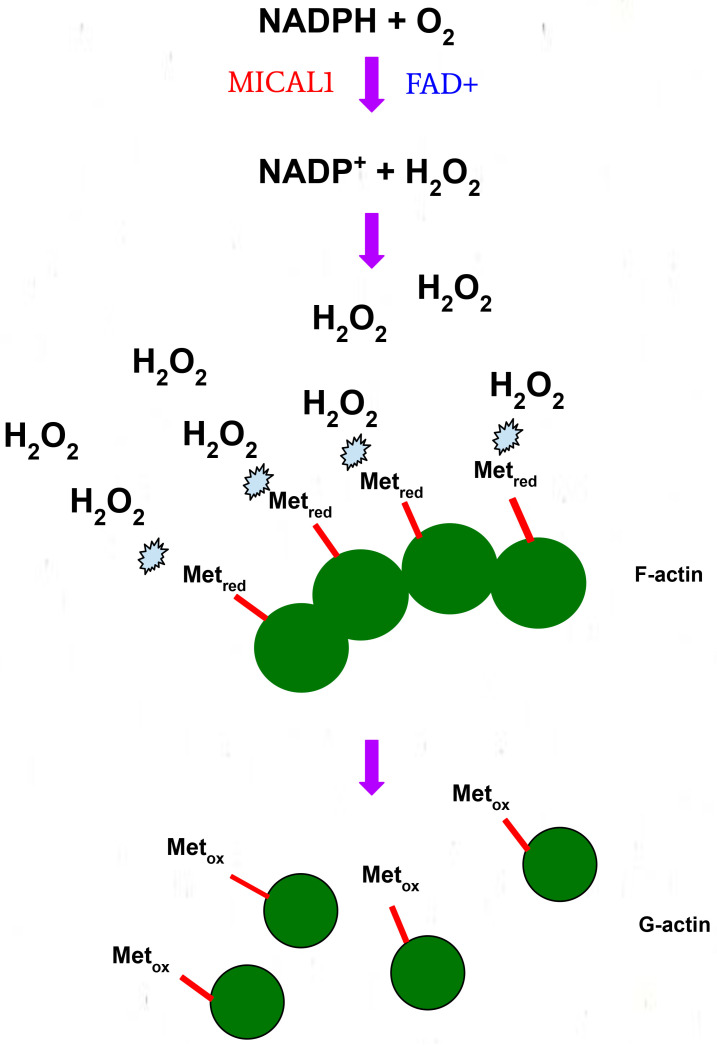
The proposed mechanism of MICAL1-mediated filamentous actin (F-actin) disassembly via the oxidation of methionine residues.

**Figure 5 genes-13-00715-f005:**
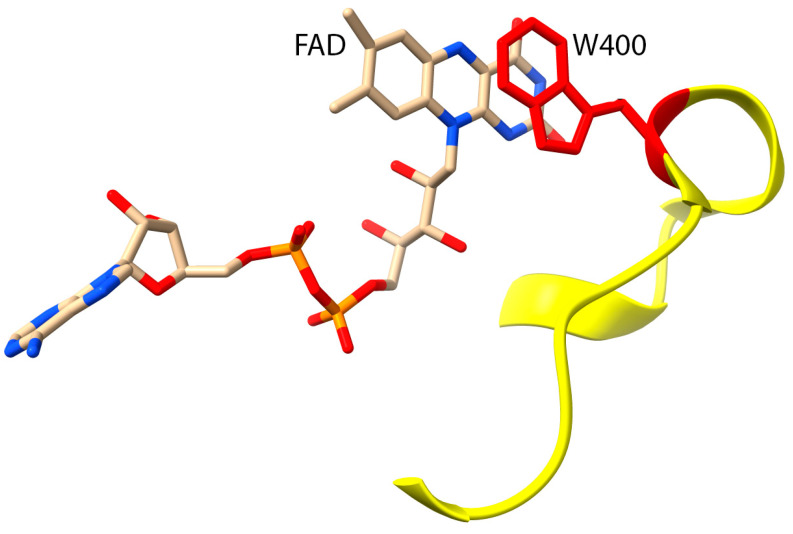
A conserved catalytic loop of MICAL1 (Residues 395–405, indicated in yellow) showing the proximity of Trp400 (W400, red) and the isoalloxazine ring of the Flavin Adenine Dinucleotide (FAD) molecule. MICAL1 protein structure was generated using UCSF Chimera (PDB 2BRA).

**Figure 6 genes-13-00715-f006:**
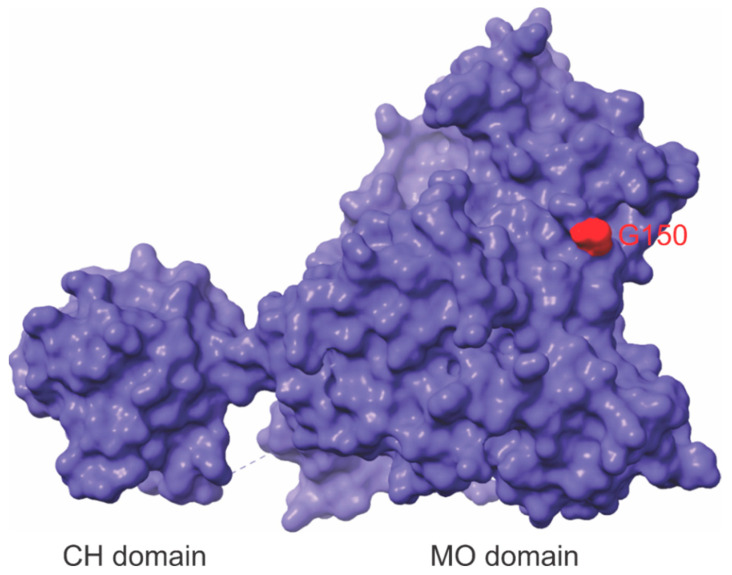
The location of G150 in MICAL1 Monooxygenase (MO) domain. The surface depiction of the MICAL1 Calponin Homology (CH) and MO domain structures (amino acids 7-613), with G150 indicated in red. The protein structure was generated using UCSF Chimera (PDB 4TXK).

**Figure 7 genes-13-00715-f007:**
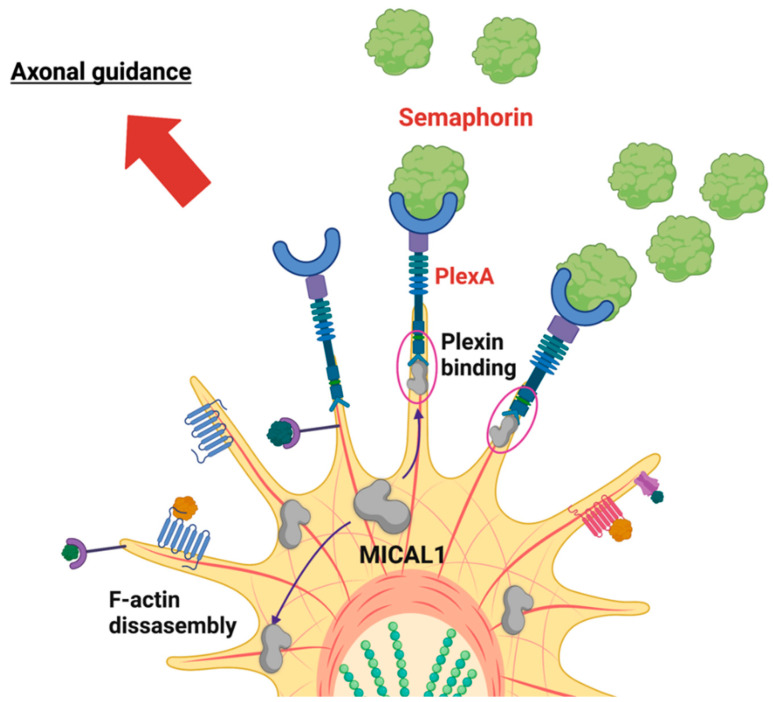
MICAL1 plays roles in axonal guidance through interactions with Plexin receptors. Through actin disassembly and plexin binding, MICAL1 regulates neurite outgrowth and eventual growth cone collapse. Mutations in *MICAL1* may affect this signaling pathway in ADLTE-affected individuals. Created with BioRender.com (accessed on 27 February 2022).

**Figure 8 genes-13-00715-f008:**
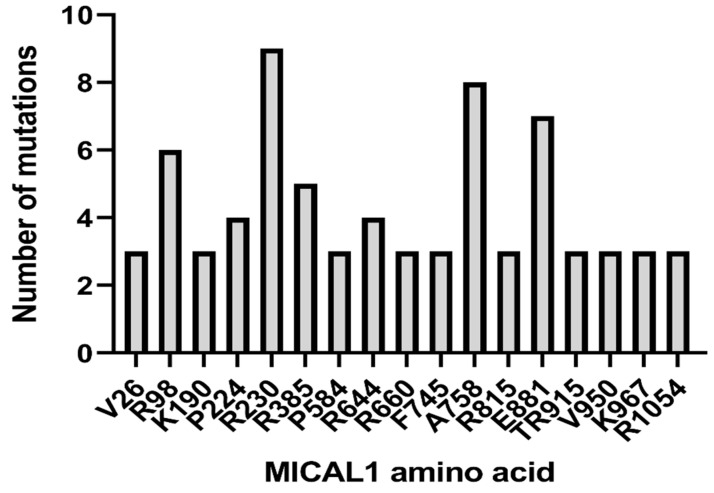
Common cancer-associated *MICAL1* mutations. *MICAL1* genomic data from 90,279 samples in 202 studies were analyzed in cBioPortal (https://www.cbioportal.org/; accessed on 1 March 2022).

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
