# Peer review of "MICAL1 Monooxygenase in Autosomal Dominant Lateral Temporal Epilepsy: Role in Cytoskeletal Regulation and Relation to Cancer"

_genes, 2022, doi:10.3390/genes13050715_

Round 1

Reviewer 1 Report

The manuscript discusses an interesting topic on MICAL1 LGI1, RELN genes, semaphorin signalling implications and possible defects in axonal function during development. Besides, it presents the possible implications of MICAL1 mutations on cancer.

I suggest minor corrections that are indicated in the attached pdf

Pag. 7: to replace lycine with glycine. Pag 8: legend of the figure: amino acids 7-613. Pag 9: WT- wild type. Page 10: to include the reference for the phrase regarding 'Copy number variations were de-347 tected in 373 tumour samples...'.

Author Response

  1. 7: to replace lycine with glycine.

RESPONSE: Corrected

  1. Pag 8: legend of the figure: amino acids 7-613.

RESPONSE: This is correct, the domains indicated in the figure represent amino acids 7 to 613

  1. Pag 9: WT- wild type.

RESPONSE: Corrected

  1. Page 10: to include the reference for the phrase regarding 'Copy number variations were de-347 tected in 373 tumour samples...'.

RESPONSE: The data in the paragraph were from cBioPortal, and the references have been added.

Reviewer 2 Report

This review describes what is currently known about the structure and function of MICAL1, and the biochemical effects of two mutations recently found to cause ADLTE, a form of familial focal epilepsy. It also discusses the possible mechanisms of cancer progression due to recurrent MICAL1 mutations, and the proposed pathogenesis of ADLTE caused by mutations of the other two genes implicated in this epileptic disorder: LGI1 and Reelin.

The paper is clearly written and illustrates in detail the known aspects of the biology of MICAL1 and the pathological effects of its alteration. Instead, the discussion of the proposed mechanisms of LGI1 mutations leading to ADLTE is oversimplified. Al least, a  couple of reviews should be cited and briefly discussed: the one about  the potential roles of the LGI1-ADAM22 complex in ADLTE (Fukata Y et al., Neuropharmacology 2021); the other discussing persistent immaturity of glutamatergic circuits due to  LGI1 mutations as a possible mechanism underlying ADLTE (Anderson MP, Epilepsy Currents 2010).  

Author Response

Reviewer 2 comments:

 Instead, the discussion of the proposed mechanisms of LGI1 mutations leading to ADLTE is oversimplified. Al least, a  couple of reviews should be cited and briefly discussed: the one about  the potential roles of the LGI1-ADAM22 complex in ADLTE (Fukata Y et al., Neuropharmacology 2021); the other discussing persistent immaturity of glutamatergic circuits due to  LGI1 mutations as a possible mechanism underlying ADLTE (Anderson MP, Epilepsy Currents 2010). 

RESPONSE: New text on lines 54-79 have elaborated on potential the roles of LGI1 mutations as causative of ADLTE, and the suggested reviews have now been cited in the revised manuscript.